# Evolving Sparsity: Leveraging Token Importance Dynamics for Efficient LLM Decoding with Sparse Attention

## Abstract

Efficient long-context inference remains a major challenge for large language models (LLMs), as the cost of attention computation during auto-regressive decoding grows linearly with the context length. Recent sparse attention methods attempt to reduce the computational burden by selecting a subset of tokens at each step, while most rely on static importance scores that are repeatedly computed over the entire cache, overlooking the relational dynamics of the decoding process. In this work, we revisit sparse attention in LLMs and propose to model token importance as a dynamic process that evolves over decoding steps and propagates through model layers. To efficiently measure token importance, we propose two lightweight mechanisms: (i) Cross-Layer Propagation, which leverages the model's intrinsic retrieval heads to compute query-aware indices and efficiently propagate them across layers; and (ii) Cross-Step Accumulation, which incrementally maintains long-term, query-agnostic importance via decayed accumulation of sparse attention scores, avoiding recomputing the importance of decoded tokens. Together, these mechanisms preserve both stable context memory and adaptive query relevance while reduce redundant computation. We evaluate our approach on PG-19, Needle-in-a-Haystack, and LongBench with models employing Multi-Head and Grouped-Query Attention. Under varying KV cache budgets, our method consistently outperforms prior sparse attention baselines, approaches full attention performance in most settings, and achieves speedups of up to $4.87\times$ for attention latency and $2.36\times$ for end-to-end decoding. Anonymous code link.

## 1 Introduction

Large Language Models (LLMs) such as GPT (Achiam et al. (2023)), LLaMA (Grattafiori et al. (2024)), and Gemini (Team et al. (2023)) have demonstrated remarkable capabilities in reasoning, knowledge retrieval, and generation across a wide range of tasks (Bai et al. (2023); Shaham et al. (2023); An et al. (2023); Zhang et al. (2024a)). A key factor enabling these abilities is the model's capacity to process long sequences of tokens. However, as context lengths grow, the computational and memory demands of the attention mechanism scale quadratically in standard architectures, creating a substantial bottleneck for long-context modeling (Fu (2024)). The challenge is further amplified during auto-regressive decoding, where each new query token requires attending over the entire preceding context.

Fortunately, LLMs exhibit a form of inherent sparsity (Deng et al. (2024)): only a subset of tokens typically contributes meaningfully to attention outputs. Motivated by this, a range of decoding-focused sparse attention techniques have been developed to reduce computation by selectively attending to likely relevant tokens rather than the entire context. Representative examples include $H_2O$ (Zhang et al. (2023)), which aggregates attention scores to surface critical tokens; StreamingLLM (Xiao et al. (2023)), which diagnoses the "attention sink" phenomenon to prioritize computation; Quest (Tang et al. (2024)), which performs query-aware, block-level evaluations during decoding. These approaches demonstrate that carefully designed sparsity can substantially lower decoding cost while preserving most useful context. Despite their differences, many of these methods still face common limitations. In particular, selection is often treated as a largely stateless, from-scratch operation at each step, without effectively leveraging signals across layers or across time—a lim-

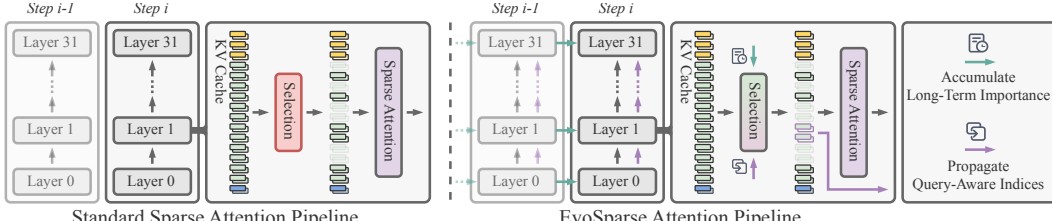

Figure 1: Pipeline overview comparing standard sparse attention with EvoSparse. Standard methods select tokens independently at each step and layer, incurring extra computation and limited selection quality. EvoSparse instead reuses query-aware indices and accumulates long-term importance across the network, leveraging cross-layer and cross-step signals to improve both effectiveness and efficiency.

itation we visualize in Figure 1. This lack of cross-layer or cross-step continuity can reduce the effectiveness of token selection, while repeated selection procedures introduce additional efficiency overheads. As a result, there remains considerable room to improve the trade-off between performance and efficiency.

Based on this observation, we propose **EvoSparse**, a framework that models token importance as a continuously evolving process (Figure 1, right). Our approach is instantiated through two complementary and efficient mechanisms: **(1) Cross-Layer Propagation**: We design a Retrieval mechanism that explicitly computes query-aware indices at a few retrieval heads (Wu et al. (2024)) and propagates these indices to non-retrieval heads in subsequent layers. This propagation guides non-retrieval heads to focus on query-aware retrieval signals while reusing the same indices, thereby reducing the computational overhead of repeated selection. **(2) Cross-Step Accumulation**: We introduce a Heat mechanism that captures long-term, query-agnostic importance. It performs a temporal decay accumulation on the sparse attention scores that are already being computed, incrementally updating a long-term relevance signal. Together, these mechanisms convert token selection into a dynamic, evolving process that improves the effectiveness of selected tokens while reducing redundant computation, yielding a better performance–efficiency trade-off.

We conduct a comprehensive evaluation on diverse long-context benchmarks (Perplexity on PG-19 (Rae et al. (2019)), Needle-in-a-Haystack (Kamradt (2023)), and LongBench (Bai et al. (2023))) with both Multi-Head Attention (Vaswani et al. (2017)) and Grouped-Query Attention (Ainslie et al. (2023)) Model. Our method achieves the best balance between performance and efficiency: it consistently outperforms existing sparse attention baselines while recovering or closely matching full attention performance under constrained KV cache budgets. Moreover, our optimized implementation delivers substantial speedups, with attention latency improved by up to $4.87\times$ and end-to-end latency by up to $2.36\times$.

## 2 RELATED WORK

**Long-Context LLMs** Efficiently handling long-context inputs is a critical challenge for modern large language models (LLMs). As context lengths grow, the memory and computation required for attention scale rapidly, creating significant bottlenecks in both training and inference. To address this, prior work has explored several complementary strategies. One approach focuses on training and fine-tuning with long-text corpora, enabling models to better capture dependencies across extended (Chen et al. (2023); Xiong et al. (2023); Fu et al. (2024)). Another line of work investigates positional encoding schemes, such as rotary positional embeddings (Su et al. (2024)) and its variants (Zhang et al. (2024b); Ding et al. (2024); Peng et al. (2023)), to support longer effective context lengths. Additionally, external memory and retrieval-augmented architectures have been proposed to provide LLMs with access to relevant long-range information without overloading the KV cache (Tworkowski et al. (2023); Mohtashami & Jaggi (2023); Xu et al. (2023)). Despite these advances, processing very long contexts remains computationally intensive, motivating more targeted methods for reducing attention cost.

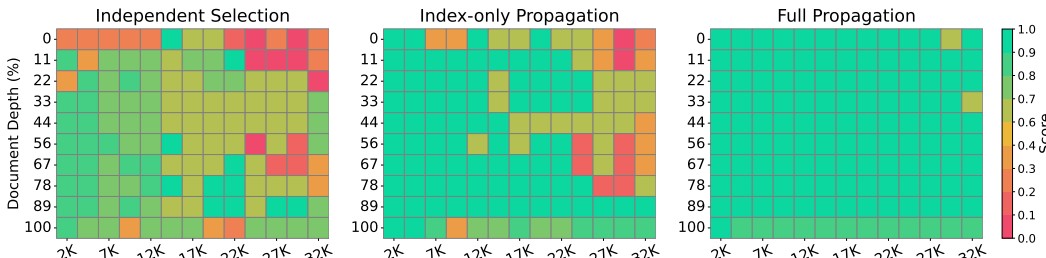

Figure 2: Effect of propagating retrieval indices to non-retrieval heads on NIHA (Kamradt (2023)) tasks with a token budget of 512. We follow Wu et al. (2024) to select the 15 heads with the highest retrieval scores as retrieval heads. Index-only Propagation substantially improves performance over Independent Selection.

**Sparse Attention** Sparse attention techniques reduce the computational cost of long-context LLMs by selecting the most relevant tokens at each decoding step. They generally fall into two categories: eviction-based methods, which remove less important tokens from the KV cache at the risk of losing information, and selection-based methods, which retain the full KV cache but dynamically focus attention on tokens most likely to influence the current output. Recent methods within this framework, including $H_2O$ (Zhang et al. (2023)), StreamingLLM (Xiao et al. (2023)), Quest (Tang et al. (2024)), Sparq Attention (Ribar et al. (2023)), Loki (Singhania et al. (2024)) and DuoAttention (Xiao et al. (2024)), introduce mechanisms such as attention score accumulation, query-aware evaluation, low-rank selection, and retrieval head to further improve efficiency while preserving essential context.

Among selection-based methods, TidalDecode (Yang et al. (2024)) employs two designated layers (selection and re-selection) to identify tokens whose indices are propagated forward, motivated by the observation of Position Persistent Sparse Attention (PPSA). While this resembles our Cross-Layer Propagation in reusing token indices, two key differences remain. First, TidalDecode exploits PPSA, whereas our method leverages retrieval heads to guide non-retrieval heads toward query-aware tokens. Second, our approach depends on inherent model signals rather than manually tuned layer-specific hyperparameters, making it more naturally integrated with existing attention structures while efficiently capturing query-aware information.

Collectively, these approaches demonstrate that decoding-phase sparsity can balance efficiency and performance, motivating our modeling of token importance as a process that accumulates over steps and propagates across layers.

## 3 METHOD

Our method is driven by two core hypotheses designed to maximize information reuse. First, we posit that retrieval indices are a transferable resource. While retrieval heads are essential for identifying query-relevant tokens from a long context (Wu et al. (2024); (Xiao et al. (2024))), the positional indices they compute are valuable in their own right. We hypothesize that the retrieval indices computed by retrieval heads can directly guide non-retrieval heads, maintaining access to query-relevant tokens across layers. To verify this, we conduct a preliminary experiment (Figure 2) comparing three configurations:

- **Independent Selection**: Retrieval heads are fully disabled. They neither provide indices nor perform attention. Each non-retrieval head independently selects its own top-k tokens and attends only to those tokens.

- **Index-only Propagation**: Retrieval heads first compute the top-k retrieval indices, which are then passed to non-retrieval heads in the same layer and in subsequent layers, while the retrieval heads themselves mask out their own attention.

- **Full Propagation**: Retrieval heads propagate their top-k indices to non-retrieval heads, but do not mask their own attention.

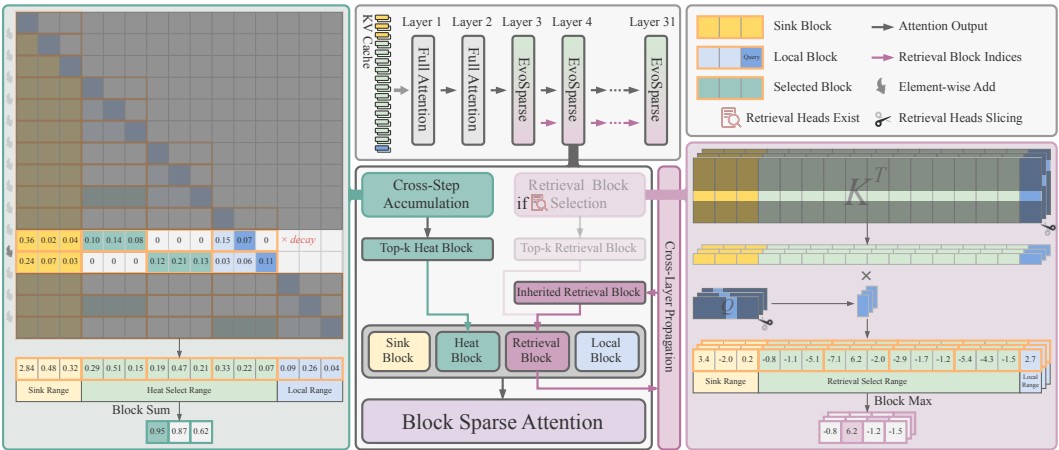

Figure 3: Overview of EvoSparse. The framework integrates two key mechanisms: (1) *Cross-Step Accumulation*, which aggregates token importance signals over time to capture long-term context; and (2) *Cross-Layer Propagation*, which reuses retrieval indices to guide non-retrieval heads, thereby enhancing their focus while maintaining efficiency.

Results show that Index-only Propagation substantially outperforms Independent Selection, though it does not fully match the performance of Full Retrieval Propagation. This demonstrates that propagating retrieval indices across layers is a powerful mechanism for maintaining focus on query-relevant information in non-retrieval heads.

Second, prior work such as $H_2O$ (Zhang et al. (2023)) suggests that token salience may exhibit temporal continuity: accumulating full attention scores can reveal query-agnostic importance maps, indicating that attention to certain tokens can persist over time. However, this approach requires access to all token attention scores, which is infeasible under sparse attention. This raises the question of whether accumulating only the sparse attention scores available at each step can similarly capture long-term token importance. We hypothesize that such step-wise accumulation may form a lightweight, persistent memory of globally salient tokens, serving as a query-agnostic prior to guide attention in subsequent steps.

## 3.1 CROSS-LAYER PROPAGATION: EXPLOITING TRANSFERABLE INDICES

To investigate whether retrieval indices can serve as a transferable resource, we introduce a Cross-Layer Propagation mechanism (Figure 3, right). This technique allows the query-aware contextual cues identified by a few retrieval heads in one layer to be efficiently shared with non-retrieval heads in subsequent layers.

The process begins in a layer $l$ containing $H$ retrieval heads. For a given query $q^l$, we compute its dot-product similarity against all keys $k^l$ in the sequence to get pre-softmax scores:

$$a_j^l = \frac{q^l (k_j^l)^\top}{\sqrt{d}}, \quad j = 1, \ldots, H. \tag{1}$$

Instead of every head performing this costly operation, the retrieval heads identify the indices of the top-$k_r$ keys with the highest scores. This creates a compact set of query-aware candidate indices:

$$\mathcal{I}_{\text{retrieval}}^l = \text{TopK}\left(\{a_j^l\}_{j=1}^H, k_r\right). \tag{2}$$

These valuable indices are then passed down the network. A subsequent layer $l + 1$ inherits these indices directly if it does not contain its own retrieval heads. This cascading behavior is defined as:

$$\mathcal{I}_{\text{retrieval}}^{l+1} = \begin{cases} \mathcal{I}_{\text{retrieval}}^l & \text{if layer } l+1 \text{ has no retrieval head,} \\ \text{TopK}((q^{l+1})(K^{l+1})^\top, k_r) & \text{if otherwise.} \end{cases} \tag{3}$$

This propagation strategy ensures that most layers receive high-quality, query-relevant indices without incurring the computational cost of a full attention score calculation, effectively democratizing the insights of the specialized retrieval heads.

## 3.2 CROSS-STEP ACCUMULATION: CAPTURING TEMPORAL SALIENCE

To explore whether token salience can be maintained over time under sparse attention, we introduce a lightweight Cross-Step Accumulation mechanism (Figure 3, left). This counters the query-aware nature of retrieval by building a persistent, query-agnostic map of token importance over the entire decoding process.

We maintain a score for each token, which we term its "Heat". At each decoding step $t$, the Heat value $h_i^t$ for every token $i$ is updated using the newly computed sparse attention scores. Let $\mathcal{I}_{\text{sparse}}^t$ be the set of indices attended to at step $t$, and $\{s_i^t\}_{i \in \mathcal{I}_{\text{sparse}}^t}$ their corresponding post-softmax attention scores. The Heat is updated via an exponential moving average:

$$h_i^t = \lambda \cdot h_i^{t-1} + s_i^t, \quad \forall i \in \{1, \ldots, N\}, \tag{4}$$

where $N$ denotes the sequence length. Here, $\lambda \in (0, 1)$ is a decay factor that gracefully reduces the influence of older scores, preventing early tokens from dominating indefinitely. For any token $i \notin \mathcal{I}_{\text{sparse}}^t$ its score $s_i^t$ is treated as zero for the update. This accumulated Heat provides a robust, long-term signal of a token's overall importance. From this, we select a candidate set of historically salient tokens:

$$\mathcal{I}_{\text{heat}}^t = \text{TopK}\left(\{h_i^t\}_{i=1}^N, \text{k}_{\text{h}}\right), \tag{5}$$

where $\text{k}_{\text{h}}$ denotes the number of top-scoring tokens retained by the heat mechanism. This mechanism provides a stable, global view of the context with minimal computational overhead, complementing the immediate, query-aware view from Cross-Layer Propagation.

## 3.3 UNIFIED SPARSE ATTENTION

At any decoding step $t$ and layer $l$, our two mechanisms produce complementary sets of indices. The final sparse attention pattern is computed over the union of these sets, along with standard sink tokens $\mathcal{I}_{\text{sink}}$ and local tokens $\mathcal{I}_{\text{local}}$:

$$\mathcal{I}^{t,l} = \mathcal{I}_{\text{sink}} \cup \mathcal{I}_{\text{retrieval}}^l \cup \mathcal{I}_{\text{heat}}^t \cup \mathcal{I}_{\text{local}}. \tag{6}$$

The attention function is then applied exclusively to the keys and values corresponding to this unified index set $\mathcal{I}^{t,l}$:

$$\text{Attn}(q^l, K^l, V^l) = \text{softmax}\left(\frac{q^l(K^l[\mathcal{I}^{t,l}])^\top}{\sqrt{d}}\right) V^l[\mathcal{I}^{t,l}]. \tag{7}$$

This composite strategy creates a powerful synergy. Cross-Layer Propagation injects immediate, query-aware relevance, while Cross-Step Accumulation provides long-term stability and coherence. Together, they enable the model to perform efficient and effective attention over long contexts, preserving performance while reducing computational requirements

## 4 EXPERIMENTS

### 4.1 SETUPS

**Tasks, Models and Baselines** We evaluate our EvoSparse on three representative long-context benchmarks: PG-19 (Rae et al. (2019)) for long-text language modeling, Needle-in-a-Haystack (NIHA) (Kamradt (2023)) for factual retrieval accuracy, and 10 tasks from LongBench (Bai et al. (2023)), including multi-document QA, single-document QA, summarization, few-shot tasks, synthetic tasks, and code-related tasks. Experiments are conducted on two large language models with distinct attention mechanisms: Llama-2-7B-32K-Instruct with standard Multi-Head Attention (Vaswani et al. (2017)), and Llama-3-8B-Instruct-Gradient-1048k (Pekelis et al. (2024)) with Grouped-Query Attention (Ainslie et al. (2023)). We compare against several training-free sparse attention baselines, including Quest (Tang et al. (2024)), TidalDecode (Yang et al. (2024)), and StreamingLLM (Xiao et al. (2023)), with Full Attention included as an upper bound.

**Implementation Details** Retrieval heads are detected following Wu et al. (2024) with a maximum sequence length of 5,000 on NIHA. For all sparse attention baselines, the first two layers remain full attention, and sparsity is applied only to subsequent layers, following Quest and TidalDecode.

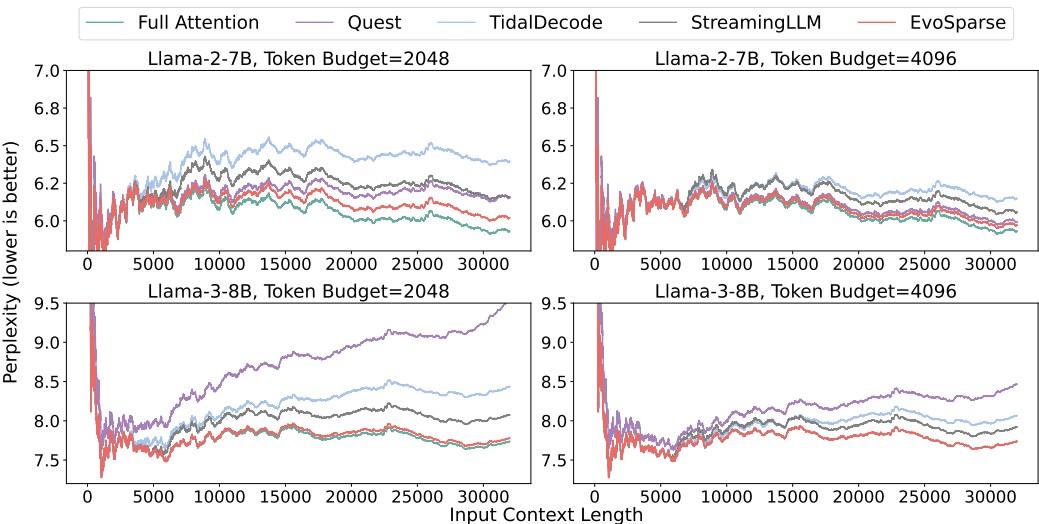

Figure 4: Perplexity of different methods across varying context lengths from 0 to 32k tokens. The results illustrate how each method scales with context length, highlighting the effectiveness of EvoSparse in maintaining low perplexity for long inputs.

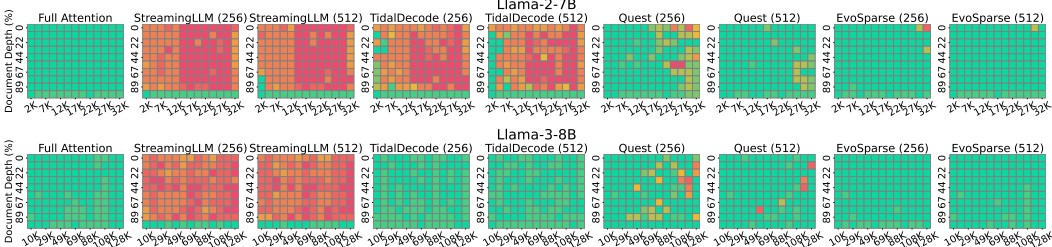

Figure 5: Performance on Needle-in-a-Haystac (NIHA) tasks across different context lengths. Llama-2 results are shown up to 32k tokens, while Llama-3 results extend to 128k tokens, illustrating how EvoSparse maintains strong retrieval accuracy in long-context scenarios.

## 4.2 PERFORMANCE EVALUATION

### 4.2.1 LANGUAGE MODELING ON PG-19

Perplexity (PPL) measures a model's ability to predict the next token, with lower values indicating stronger language modeling. As shown in Figure 4, EvoSparse achieves PPL comparable to Full Attention across models and token budgets, preserving performance under constrained conditions. In contrast, baselines exhibit limitations: Quest retrieves across all heads, introducing noise from ineffective ones; TidalDecode is sensitive to re-selection layer, performing well on LLaMA-3 but dropping on LLaMA-2 when layers mismatch; StreamingLLM drops tokens indiscriminately, causing consistent PPL degradation.

### 4.2.2 FACTUAL RETRIEVAL ON NEEDLE-IN-A-HAYSTACK

Needle-in-a-Haystack (NIHA) (Kamradt (2023)) evaluates a model's ability to retrieve relevant tokens from extremely long contexts under limited token budgets. As shown in Figure 5, baseline methods exhibit various limitations: StreamingLLM degrades due to fixed token eviction; Quest struggles under small budgets as retrieval across all heads introduces noise, with only 1–5% of heads being effective retrieval heads (Wu et al. (2024)). TidalDecode performs reasonably well on Llama-3, but its performance drops markedly on Llama-2. We note that TidalDecode is highly sensitive to the choice of re-selection layer, which requires manual tuning per model. While prior work reports the optimal layer to be Layer 13 for Llama-3, we follow their setting on Yarn-Llama-2-7B-128K

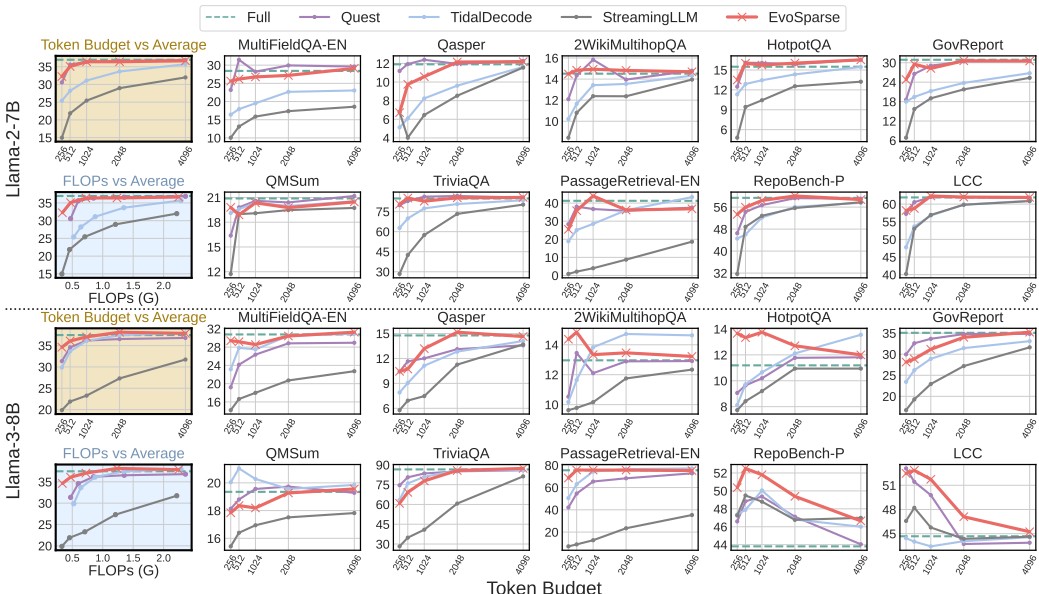

Figure 6: Evaluation on 10 LongBench tasks under varying token budgets {256, 512, 1024, 2048, 4096}. For each task, individual performance scores are reported, along with the average score across all tasks at each budget. We additionally analyze the relationship between average score and attention FLOPs, highlighting the efficiency-performance trade-off of different methods.

(Peng et al. (2023)) and adopt Layer 7 for Llama-2. This mismatch partly explains the performance degradation observed on Llama-2. By contrast, our method achieves nearly identical retrieval accuracy to full attention across, under both 256 and 512 token budgets. This demonstrates that our approach robustly preserves retrieval capability without requiring model-specific or layer-specific tuning, consistently outperforming all baselines.

### 4.2.3 GENERAL LONG-CONTEXT CAPABILITIES ON LONGBENCH

LongBench (Bai et al. (2023)) is a diverse suite of long-context benchmarks spanning multiple domains, designed to evaluate a model's ability to handle extended inputs across retrieval, question answering, and summarization. We report results on 10 representative LongBench tasks, as shown in Figure 6. In addition to per-task scores, we provide averaged performance under different token budgets, as well as the attention computation cost per forward pass at the average sequence length of these tasks, measured in terms of matrix multiplications within the attention module (excluding the Q/K/V/O projections).

From an overall perspective, our method achieves the best average performance across both models compared to StreamingLLM, TidalDecode, and Quest. At higher token budgets, our method attains accuracy on par with full attention, while at lower token budgets, it surpasses all baselines. StreamingLLM suffers from its fixed token dropping strategy, Quest is less effective under small budgets due to noisy retrieval across all heads, and TidalDecode performs relatively well on Llama-3 but falls behind on Llama-2 because of its reliance on manually tuned re-selection layers. In contrast, our approach maintains strong retrieval accuracy while reducing attention computation, showing robustness across architectures and budget regimes.

### 4.3 EFFICIENCY EVALUATION

### 4.3.1 THEORETICAL FLOPs ANALYSIS

We analyze the theoretical FLOPs of different attention mechanisms under the decoding setting. Let $N$ denote the sequence length in the KV cache, $d$ the hidden dimension per head, $k$ the number of tokens selected by sparse attention, and $B$ the block size in block-based strategies.

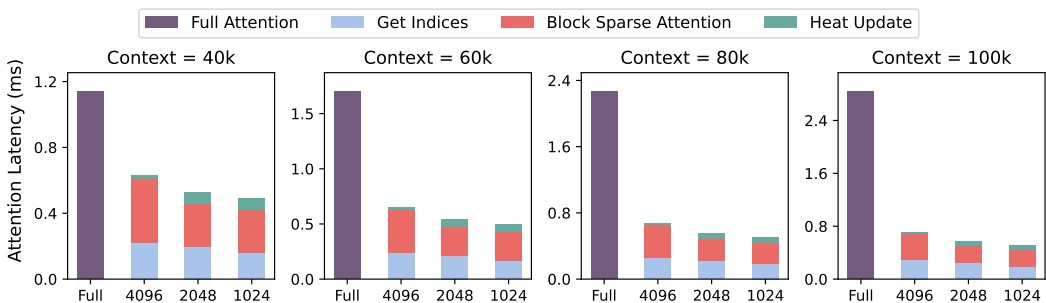

Figure 7: Attention latency of EvoSparse components under different token budgets {1024, 2048, 4096} across context lengths from 40k to 100k tokens. The results illustrate the contribution of each component to overall latency and demonstrate the scalability of EvoSparse in long-context scenarios.

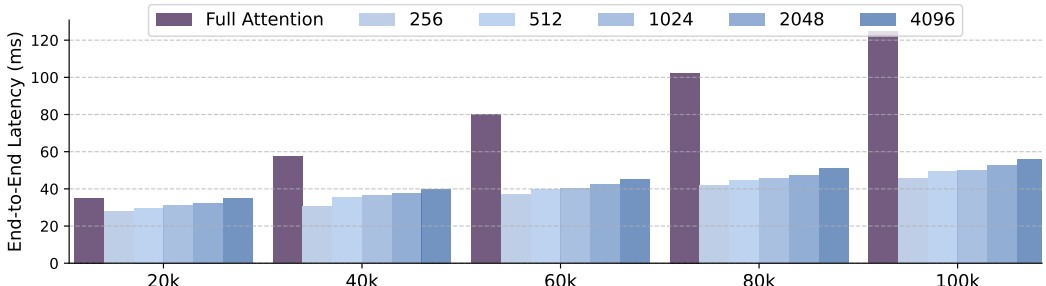

Figure 8: End-to-end latency of EvoSparse under varying token budgets {256, 512, 1024, 2048, 4096} across context lengths from 20k to 100k tokens. The results demonstrate how overall inference time scales with both token budget and context length, highlighting the efficiency of EvoSparse in long-context scenarios.

**Full Attention** Each head requires $4dN$ FLOPs to compute $(QK^\top)V$.

**Quest** (Tang et al. (2024)) Each head consumes $4dk + 3d\lceil N/B \rceil$ FLOPs, as it performs retrieval across all heads irrespective of their contribution, which results in overhead under small token budgets.

**TidalDecode** (Yang et al. (2024)) Non-retrieval head cost $4dk$ FLOPs, while heads in the selection layer incur $4dN$ FLOPs.

**StreamingLLM** (Xiao et al. (2023)) Each head requires $4dk$ FLOPs.

**EvoSparse** Non-retrieval heads incur $4dk$ FLOPs, while retrieval heads require $4dk + 2dN$ FLOPs. In addition, heat computation and update introduce $2N - \lceil N/B \rceil$ FLOPs operations per layer.

By considering model depth, head dimension, number of heads, and average sequence lengths in LongBench, we estimate the attention FLOPs for a single forward pass of each method. As shown in Figure 6, EvoSparse achieves the best trade-off between attention FLOPs and average score, reflecting the effect of extensive reuse of information across steps and layers.

### 4.3.2 EMPIRICAL LATENCY EVALUATION

We further benchmark the empirical efficiency of EvoSparse on Llama-3-8B-Instruct-Gradient-1048k (Pekelis et al. (2024)) using an RTX 5090 GPU with BF16 precision. We report the results for two aspects of latency under varying context lengths and token budgets: the attention latency alone (Figure 7) and the end-to-end decoding latency for generating a single token (Figure 8).

In particular, under a 100K context with a 2048-token budget, EvoSparse accelerates the attention computation by up to 4.87× relative to Full Attention. This improvement directly translates to end-to-end decoding: EvoSparse yields up to 2.36× speedup under the same setting.



Figure 9: Ablation study on NIHA tasks using Llama-2-7B-32K-Instruct. The results highlight the contribution of each component to retrieval accuracy.

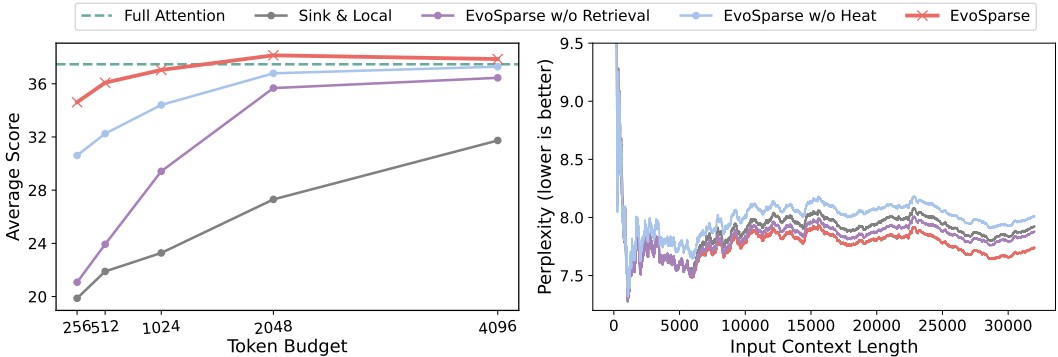

Figure 10: Ablation study on both LongBench tasks and Perplexity using Llama-3-8B-Instruct-Gradient-1048k. The results highlight the contribution of each component across language modeling and long-context comprehensive benchmarks.

## 4.4 ABLATION STUDY

To quantify the contribution of EvoSparse's components, we perform ablation experiments on PG-19 Perplexity, NIHA, and 10 tasks from LongBench. EvoSparse consists of a *sink & local* streaming backbone, a *heat* mechanism preserving long-term token importance, and an explicit *retrieval* mechanism reusing selected retrieval tokens. We compare the full model against two ablations (EvoSparse w/o Heat & EvoSparse w/o Retrieval) and the Sink & Local backbone as a baseline.

**Effect of the retrieval mechanism** Removing retrieval causes a clear accuracy drop on NIAH (Figure 9) and LongBench (Figure 10, left), but only moderately affects PG-19 perplexity (Figure 10, right). This reflects that explicit retrieval is essential for locating sparse, task-relevant evidence, whereas heat alone cannot fully recover such information.

**Effect of the heat mechanism** Removing heat leads to the largest drop in PG-19 perplexity (Figure 10, right), while having only a moderate impact on NIAH (Figure 9) and LongBench (Figure 10, left). This indicates that heat primarily supports language-modeling quality by maintaining globally relevant context, partially compensating for the lack of explicit retrieval.

**Sink & Local backbone** The streaming backbone without heat or retrieval performs worst, showing that both mechanisms provide complementary gains for long-context modeling.

## 5 CONCLUSION

We presented EvoSparse, a simple yet effective sparse attention framework that reuses information across layers and decoding steps. By propagating transferable retrieval indices and accumulating sparse attention score, EvoSparse achieves a synergy of query-aware relevance and long-term stability. Experiments across PG-19, NIHA, and LongBench demonstrate that EvoSparse matches full attention under generous budgets while substantially outperforming prior sparse methods under constrained settings. In addition, EvoSparse delivers up to $4.9\times$ faster attention computation and $2.4\times$ faster end-to-end decoding, offering a strong efficiency-performance trade-off for long-context LLM decoding.

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

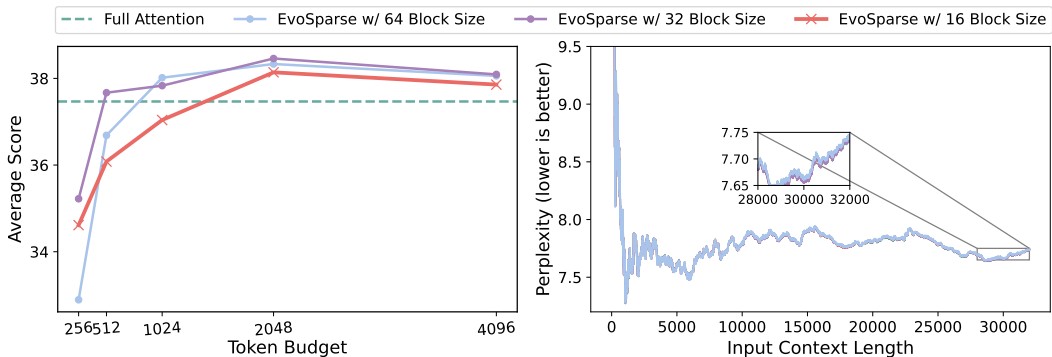

Figure 1: Ablation study on both Perplexity and LongBench tasks using Llama-3-8B-Instruct-Gradient-1048k, examining the impact of varying block sizes.

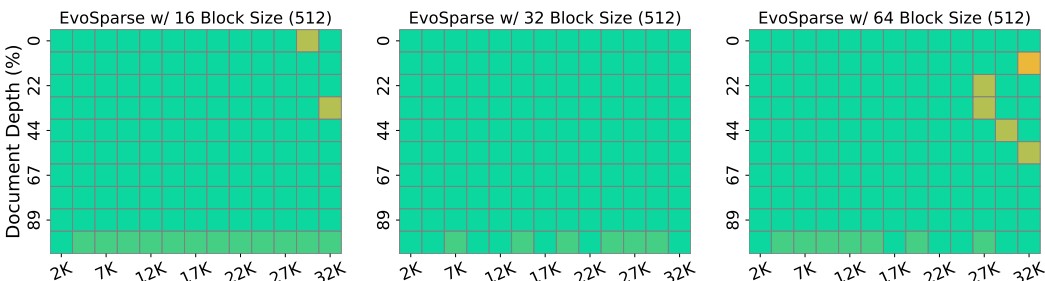

Figure 2: Ablation study on NIHA tasks using Llama-2-7B-32K-Instruct, examining the impact of varying block sizes.

## A APPENDIX

### A.1 STATEMENT ON THE USE OF LARGE LANGUAGE MODELS

In the preparation of this manuscript, we used large language models (LLMs), such as ChatGPT, solely for language polishing and improving the clarity and readability of the text. All scientific content, experimental design, data analysis, and conclusions were independently developed by the authors. The LLM was not involved in any scientific decision-making or data processing.

### A.2 ADDITIONAL ABLATION STUDIES

To further validate the design choices and robustness of EvoSparse, we conduct additional ablation studies on key hyperparameters and design decisions.

#### A.2.1 ABLATION ON BLOCK SIZE

Since EvoSparse operates at a block level, the choice of block size is an important hyperparameter. We experiment with block sizes of 16, 32, and 64. The default block size used in our main experiments is 16. The experiments are conducted on the PG-19 (Rae et al. (2019)) (Perplexity), Needle-in-a-Haystack (NIHA) (Kamradt (2023)), and LongBench (Bai et al. (2023)) benchmarks. The results are presented in Figure 1 and Figure 2.

**PG-19 Perplexity** We observe no significant difference in perplexity across the three block size (Figure 1, right). This demonstrates the robustness of EvoSparse on language modeling tasks, where performance is not highly sensitive to the granularity of token selection.

**LongBench** Interestingly, on the diverse tasks within LongBench, block sizes of 32 and 64 often perform on par with or even slightly better than the default block size of 16 (Figure 1, left). The

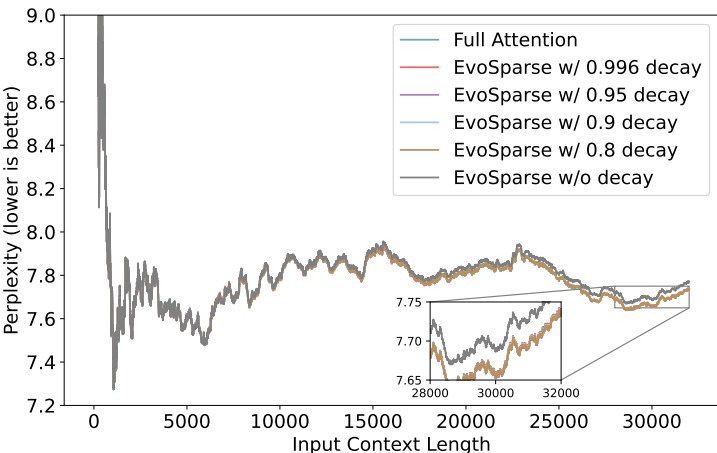

Figure 3: Ablation study on Perplexity task using Llama-3-8B-Instruct-Gradient-1048k,examining the impact of varying decay hyperparameters.

only notable exception is a slight performance drop for the 64-block size under the tightest 256-token budget. Overall, these results highlight the stability of EvoSparse across different block configurations. To ensure a fair and direct comparison with prior work like Quest (Tang et al. (2024)), which uses a block size of 16, we retain this value for all experiments reported in the main paper.

**Needle-in-a-Haystack**    The NIHA task appears to be moderately more sensitive to block size (Figure 2). The performance with a block size of 32 is nearly identical to that of 16, but we observe a slight degradation with a block size of 64. This suggests that for retrieval tasks like NIHA, a smaller block size might be more effective at precisely isolating the "needle" token.

### A.2.2    Ablation on Decay Factor $\lambda$

The Cross-Step Accumulation mechanism relies on a decay factor, $\lambda$, to balance the influence of past and present attention scores. We analyze the sensitivity of our model to this hyperparameter. We conduct this ablation on the PG-19 perplexity task, varying $\lambda$ across a range of values.

Our experiments reveal that the model's performance is highly robust to the choice of $\lambda$, with nearly identical perplexity scores for all tested values except for $\lambda=1.0$ (Figure 3). The poor performance at $\lambda=1.0$ (i.e., no decay) is expected and aligns with our analysis in Section 3.2 of the main paper. Without decay, the accumulation of sparse attention scores is susceptible to the "Matthew effect": tokens selected in early decoding steps will have their scores perpetually increased, making them more likely to be selected again. This process effectively prevents other potentially relevant tokens from ever being considered, leading to a significant degradation in modeling quality. The stability across other $\lambda$ values demonstrates that EvoSparse does not require extensive tuning of this hyperparameter.

### A.3    Discussion

### A.3.1    Further Details on Efficiency OptimizationThe

The significant speed improvements of EvoSparse reported in the main paper stem from several key implementation optimizations beyond the reduction in theoretical FLOPs.

**Block-wise KV Cache Handling**    We process the KV cache in blocks, which yields two primary benefits. First, it helps maintain the memory contiguity of the large KV cache tensor, which is crucial for efficient memory access on modern hardware. Second, our block-wise gather strategy is more efficient for loading the selected blocks into compute units compared to gathering individually selected tokens from scattered memory locations.

**Optimized repeat_kv for GQA** Grouped-Query Attention architectures require repeating the Key and Value heads to match the number of Query heads before the attention computation. In a naive implementation for long contexts, this repeat operation on the entire KV cache introduces a non-negligible latency bottleneck. Our implementation mitigates this by applying the repeat operation only to the selected, important blocks of the KV cache after they have been gathered. This dramatically reduces the size of the tensor being repeated, significantly lowering the overhead associated with this step and contributing to the overall end-to-end latency reduction.

Notably, our current implementation does not rely on specialized kernel frameworks such as Triton (Tillet et al. (2019)) or custom CUDA kernels. Despite being implemented purely in PyTorch, EvoSparse already achieves substantial inference speedups over full attention. We expect that leveraging high-performance kernel frameworks in future work could further accelerate our method and amplify these gains.

