# OpenReview forum: "Evolving Sparsity: Leveraging Token Importance Dynamics for Efficient LLM Decoding with Sparse Attention"
_ICLR.cc/2026/Conference — ICLR 2026 Conference Withdrawn Submission_

### Official Review · Reviewer_uELy · 2025-10-18

**Soundness:** 2
**Presentation:** 2
**Contribution:** 2
**Rating:** 4
**Confidence:** 3

**Summary:**

The paper introduces EvoSparse, a sparse attention framework designed to enhance the efficiency of long-context large language model (LLM) decoding. It models token importance as a dynamic process evolving across both decoding steps and model layers. Two key components are proposed:
* Cross-Layer Propagation, which reuses retrieval head indices to inform non-retrieval heads, enabling query-aware selection without recomputation.
* Cross-Step Accumulation, which maintains a decayed accumulation of sparse attention scores (termed “Heat”) to capture long-term, query-agnostic token salience.
Evaluations across PG-19, Needle-in-a-Haystack (NIHA), and LongBench benchmarks using LLaMA-2 and LLaMA-3 models show that EvoSparse achieves comparable or superior performance to full attention and existing sparse methods (Quest, TidalDecode, StreamingLLM), with up to 4.87× speedup in attention computation and 2.36× faster decoding.

**Strengths:**

1. Modeling token importance as an evolving process that integrates both cross-layer and temporal sparsity mechanisms represents a novel and insightful approach to sparse attention design.
2. The performance is good.

**Weaknesses:**

1. The paper’s hypotheses regarding token importance dynamics remain largely intuitive without sufficient formal or empirical substantiation. Including visual analyses of which tokens are retained across decoding steps would significantly improve interpretability and validate the “evolving importance” hypothesis.
2. The reuse of retrieval indices across layers may introduce cumulative bias or propagate attention errors. This potential limitation is not analyzed or experimentally measured.
3. The Cross-Step Accumulation mechanism lacks both a theoretical foundation and deeper empirical justification. The design choice of the decay factor and its effect on model stability and accuracy should be further clarified.

**Questions:**

1. Since retrieval heads are central to the proposed approach, the paper would benefit from a brief introduction to the retrieval head concept in the background section. Additionally, the Index-only Propagation experiment should include implementation details on how propagation is performed.
2. The two core hypotheses currently remain intuitive. It would strengthen the paper to provide quantitative analysis or visualization showing the percentage of token indices that remain stable across layers or steps.
3. How many retrieval heads does a typical model contain? Does this number vary with context length, dataset, or architecture?
4. The paper does not compare EvoSparse with recent efficient attention techniques such as MInference or SampleAttention. These models achieve strong efficiency in long-context prefill. It would be valuable to clarify whether index propagation provides superior token selection accuracy or latency improvements compared to these methods.
5. Evaluate EvoSparse on latest state-of-the-art LLMs to verify its generality and adaptability across architectures.
6. Conduct experiments across different domains (e.g., code, dialogue, narrative) to determine whether the optimal decay factor (λ) is task-dependent or stable across datasets.

---

### Official Review · Reviewer_TVMf · 2025-10-31

**Soundness:** 3
**Presentation:** 2
**Contribution:** 2
**Rating:** 2
**Confidence:** 4

**Summary:**

As context length increases, the decoding process becomes increasingly attention-bound.
While existing sparse attention methods attempt to reduce this cost, most of them are *stateless*, that is, each layer and each decoding step independently determines its sparsity pattern.
This paper proposes **EvoSparse**, a stateful sparse attention mechanism that (1) uses the retrieval heads of the previous layer to identify important tokens for the non-retrieval heads of the next layer, and (2) introduces a *Heat* mechanism to track the historical importance of each token.
EvoSparse aims to achieve a favorable trade-off between efficiency and accuracy.

**Strengths:**

- *EvoSparse* presents an interesting observation that the tokens selected by retrieval heads can guide the sparsity pattern of non-retrieval heads.
- The method operates effectively under a small token budget regime.

**Weaknesses:**

- The motivation for introducing a stateful sparsity pattern is not clearly articulated.
- The efficiency evaluation lacks accuracy and does not convincingly reflect real-world performance.

**Questions:**

Thanks for submitting to ICLR 2026. The paper emphasizes the potential advantages of a stateful sparsity pattern. However, I still have some concerns about the paper.

## 1. Motivation and necessity of statefulness
The motivation for moving from stateless to stateful sparsity patterns is not sufficiently justified. Not enough evidence is provided to show that existing stateless methods fail to capture important tokens or make suboptimal selections. In the preliminary study, while *index-only* and *full propagation* outperform *independent selection*, it remains unclear whether using retrieval heads alone can achieve comparable performance. The assumption that temporal salience persists and cannot be captured by stateless sparsity is also unsubstantiated. Can the authors provide concrete examples where existing methods fail to identify critical tokens?

## 2. Evaluation methodology and efficiency metrics
The efficiency analysis primarily reports FLOPs, but attention operations are typically memory-bound. FLOPs alone cannot represent the true bottleneck. A more appropriate metric would be the memory footprint or the actual runtime latency. However, even memory footprint does not directly correlate with latency, as complex sparsity mechanisms often make it difficult, or even impossible, to implement high-performance kernels. Given that *EvoSparse* involves separating retrieval and non-retrieval heads, top-k selection, Heat updates, and union operations, the actual runtime performance remains unclear.

## 3. Baselines and implementation fairness
Although the paper reports runtime results, all baselines and *EvoSparse* are implemented purely in PyTorch. The claim that "despite being implemented purely in PyTorch, EvoSparse already achieves substantial inference speedups over full attention" is not convincing. Optimizing an already well-tuned attention kernel (e.g., FlashAttention or FlashInfer) is substantially harder than optimizing naive PyTorch code. Moreover, the reported runtime on an RTX 5090 for GQA LLaMA3-8B is slower than *Quest* on an RTX 4090 for a non-GQA LLaMA2-7B model, which raises concerns about the validity of the measurements. To make the performance evaluation credible, the full attention baseline should use optimized kernels such as FlashAttention or FlashInfer, and a corresponding high-performance kernel for *EvoSparse* should be implemented.

## 4. Token budget allocation
The paper states that *EvoSparse* takes the union of sink, retrieval, head, and local tokens. Given a fixed token budget, how is the allocation among these components determined? For instance, if the total budget is 512 tokens, how many are drawn respectively from retrieval, head, and local selections to ensure the final union remains under 512?

## 5. Handling of GQA
How does *EvoSparse* handle grouped-query attention (GQA)? When multiple query heads identify different important tokens, how are these selections unified?

## 6. Missing baseline comparisons
Given that *EvoSparse* distinguishes between retrieval and non-retrieval heads, *DuoAttention* appears to be a natural baseline for comparison. Furthermore, the current benchmarks (e.g., perplexity, NIHA, and LongBench) are relatively easy long-context tasks. It would strengthen the paper to include results on more challenging benchmarks such as *RULER*.

## 7. Unexpected perplexity results
In the perplexity evaluation, *stringinLLM* surpasses some baselines. What contributes to this outcome?

## 8. Applicability of the Heat mechanism
For datasets where the model produces only a final answer (e.g., MMLU), how does the Heat mechanism function?

---

### Official Review · Reviewer_Rupx · 2025-11-01

**Soundness:** 3
**Presentation:** 2
**Contribution:** 3
**Rating:** 6
**Confidence:** 3

**Summary:**

This paper proposes EvoSparse, a framework for token selection for long-context LLMs. Existing frameworks perform token selection in a stateless manner, which causes ineffective token selections, as well as redundant computations. In contrast, EvoSparse uses two mechanisms that models token importance as a dynamic process that evolves over decoding steps and propagates through the model layers. First, EvoSparse uses cross-layer propagation, where the model’s retrieval heads compute query-aware indices and then pass this information to non-retrieval heads in the subsequent layers. This helps non-retrieval heads to focus on query-aware signals while reusing the same indices, which reduces computational overhead. .Second, EvoSparse uses cross-step accumulation, which performs a temporal decay accumulation on the sparse attention scores, which incrementally updates the long-term relevance signal.

**Strengths:**

The paper solves an important problem of achieving effective token selection for long-context LLMs which are growing in prevalence
The technique introduced to transfer information across layers to avoid redundant computation and increase effectiveness is interesting and makes sense
The overall performance in terms of efficiency and FLOPS is significant, compared to prior baselines

**Weaknesses:**

It would have been good to get a sense of the retrieval heads across layers for different models, which helps understand how much recomputation is avoided. Isn’t it true that the subsequent layers generally have more retrieval heads compared to the initial layers. It would be good to know if certain models are unsuitable depending on how the retrieval and non-retrieval heads are distributed across layers.
The writing could be improved to understand the performance better. The figures don’t have explanations, and are missing descriptions about the axes to make sure that the performance is well understood.
It is unclear how much the sensitivity is generally towards the decay factor, across models. I understand that there is some analysis for Llama-3-8B-1048K in the appendix, but not clear how this is generic across models and datasets.

**Questions:**

Please try to answer as many questions as possible from the weakness section.
Can you talk about the generality of the design when it comes to whether all models can benefit from your techniques generally, or are some trained models not suitable for your methods?
Can you talk more about the sensitivity of different parameters in different models, and how to configure those parameters in general, to understand the practicality of this work?

---

### Official Review · Reviewer_6LJo · 2025-11-03

**Soundness:** 3
**Presentation:** 3
**Contribution:** 2
**Rating:** 4
**Confidence:** 5

**Summary:**

This paper addresses the well-known latency bottleneck of long-context inference in LLMs. The core idea is to model token importance as a dynamic, evolving process, rather than a static property recomputed at each step.

The proposed EvoSparse framework is built on two complementary mechanisms. First, Cross-Layer Propagation identifies the model's intrinsic "retrieval heads," uses them to compute key token indices, and then efficiently propagates these indices to subsequent layers. This avoids redundant computation. Second, Cross-Step Accumulation (or "Heat") uses an EMA to maintain a long-term, query-agnostic importance score, tracking tokens that are consistently relevant.

The authors validate this on Llama-2/3 models using standard benchmarks (NIHA, LongBench, PG-19), reporting that their method outperforms baselines, approaches full-attention performance, and achieves a significant 2.36x end-to-end speedup.

**Strengths:**

1. Novel and Elegant Core Concept: The decoupling of importance into query-aware (propagation) and query-agnostic (accumulation) signals is a strong and intuitive idea.

2. Exceptional NIHA Performance: The method's ability to nearly match full-attention on the difficult Needle-in-a-Haystack benchmark (Figure 5) is a standout achievement, as this is a common failure point for many sparse attention techniques.

3. Practical, Measured Speedups: The paper provides empirical end-to-end latency measurements (up to 2.36x), not just theoretical FLOPs. This demonstrates real-world, practical efficiency gains.

4. Strong Ablation Study: The ablations clearly validate the design, proving that both the retrieval and heat mechanisms are necessary and complementary.

5. Methodological Robustness: The appendix shows that the method is not overly sensitive to key hyperparameter choices, which is a significant practical advantage.

**Weaknesses:**

The paper is well-executed, but its evaluation contains a critical omission that significantly limits the scope of its claims.

1. Evaluation is Limited to "Long-Input" Scenarios: The paper's benchmarks (NIHA, LongBench) almost exclusively test the case where the input prompt is long, but the generated output is relatively short. The paper fails to evaluate the "long-generation" workload (e.g., long Chain-of-Thought), where the KV cache grows dynamically over thousands of steps.

2. Applicability to Reasoning is Untested and Unclear: This is the central weakness. Long CoT reasoning is a primary bottleneck for LLMs. It is completely unknown how EvoSparse's mechanisms would behave in this scenario. For instance, how does the "Heat" accumulator perform over thousands of generation steps? Could it erroneously "latch on" to early tokens and fail to adapt? How does the "Retrieval" mechanism function when attending to the model's own generated content? The lack of any data on this is a major gap.

**Questions:**

1.  Can the authors provide any data on "long-generation" tasks? I am specifically concerned about the stability and effectiveness of the "Heat" and "Retrieval" mechanisms during rollouts spanning thousands of tokens (e.g., long CoT or summarization).
2.  Following up on this, have you considered evaluating EvoSparse on models specifically designed for reasoning (e.g., DeepSeek-R1 or Qwen3 "Thinking" models)? It would be crucial to see if acceleration can be achieved *without* degrading the model's complex reasoning capabilities.
3.  Could you elaborate on the retrieval head dependency? How many are typically identified in the tested models? What is the proposed alternative for the "Cross-Layer Propagation" mechanism if a model does not exhibit this property?
4.  Please clarify the budget allocation. For a given token budget (e.g., 2048), how are the sizes of the $\mathcal{I}_{local}$, $\mathcal{I}_{heat}$, and $\mathcal{I}_{retrieval}$ sets determined? Are $k_h$ and $k_r$ fixed hyperparameters, or are they a dynamic fraction of the total budget?

---

### Note · Authors · 2025-12-01

I have read and agree with the venue's withdrawal policy on behalf of myself and my co-authors.